# Cathelicidin Antimicrobial Peptide Levels in Atherosclerosis and Myocardial Infarction in Mice and Human

**DOI:** 10.3390/ijms25052909

**Published:** 2024-03-02

**Authors:** Alexandra Höpfinger, Andreas Schmid, Thomas Karrasch, Sabine Pankuweit, Andreas Schäffler, Karsten Grote

**Affiliations:** 1Department of Internal Medicine III, University of Giessen, Klinikstr. 33, 35392 Giessen, Germany; andreas.schmid@innere.med.uni-giessen.de (A.S.); thomas.karrasch@innere.med.uni-giessen.de (T.K.); andreas.schaeffler@innere.med.uni-giessen.de (A.S.); 2Department of Cardiology and Angiology, Philipps-University Marburg, Baldinger Str., 35043 Marburg, Germany; pankuwei@staff.uni-marburg.de (S.P.); karsten.grote@staff.uni-marburg.de (K.G.)

**Keywords:** cathelicidin antimicrobial peptide, atherosclerosis, coronary artery disease, Ldlr^−/−^ mice

## Abstract

Obesity represents a worldwide health challenge, and the condition is accompanied by elevated risk of cardiovascular diseases caused by metabolic dysfunction and proinflammatory adipokines. Among those, the immune-modulatory cathelicidin antimicrobial peptide (human: CAMP; murine: CRAMP) might contribute to the interaction of the innate immune system and metabolism in these settings. We investigated systemic CAMP/CRAMP levels in experimental murine models of atherosclerosis, myocardial infarction and cardiovascular patients. Atherosclerosis was induced in low-density lipoprotein receptor-deficient (Ldlr^−/−^) mice by high-fat diet (HFD). C57BL/6J wild-type mice were subjected to myocardial infarction by permanent or transient left anterior descending (LAD)-ligation. *Cramp* gene expression in murine organs and tissues was investigated via real-time PCR. Blood samples of 234 adult individuals with or without coronary artery disease (CAD) were collected. Human and murine CAMP/CRAMP serum levels were quantified by ELISA. Atherosclerotic mice exhibited significantly increased CRAMP serum levels and induced *Cramp* gene expression in the spleen and liver, whereas experimental myocardial infarction substantially decreased CRAMP serum levels. Human CAMP serum quantities were not significantly affected by CAD while being correlated with leukocytes and pro-inflammatory cytokines. Our data show an influence of cathelicidin in experimental atherosclerosis, myocardial infarction, as well as in patients with CAD. Further studies are needed to elucidate the pathophysiological mechanism.

## 1. Introduction

Coronary artery disease (CAD) is one of the main causes of death in high-income countries [1]. The incidence of CAD has been increasing year-to-year since 1987 [2]. Therefore, research on mechanisms underlying the pathogenesis of CAD is highly relevant and the focus of numerous current studies. CAD is mainly caused by coronary arterial stenosis, resulting in ischemia and cell death of cardiac muscle [3]. Known risk factors include arterial hypertension, dyslipidaemia, and obesity [3], which are associated with low-grade systemic inflammation [4]. In the adipose tissue of obese individuals, the infiltration of activated, proinflammatory monocytes/macrophages is increased, leading to a local and, hereinafter, systemic cascade of inflammation [4].

Cathelicidins are a family of antimicrobial peptides and are a part of the innate immune system [5]. About 30 cathelicidin family members have been identified in mammalian species [5]. The 37-amino acid-long peptide LL-37 (cathelicidin antimicrobial peptide: CAMP) is the only cathelicidin-derived antimicrobial peptide found in humans [5,6]. The homologue peptide in mice is called mouse cathelicidin-related antimicrobial peptide (CRAMP) [5,7,8,9]. CRAMP shows striking similarities to CAMP, creating a useful model for study in regard to human cathelicidin function and regulation [10]. Cathelicidins are mainly produced by immune cells, such as monocytes, lymphocytes, or natural killer cells [11]. Cathelicidins play an important role in host defense, e.g., they can induce the degranulation of granulocytes [12] and perforate bacterial cell membranes [13]. Cathelicidins also act as a chemoattractant on monocytes and granulocytes [14]. In 2015, Zhang et al. discovered that cathelicidin is produced by murine adipocytes and that its functional expression in subcutaneous adipose tissue is crucial for sufficient host defense in gram-positive infection [15]. Recently, CAMP was described to play a role in atherosclerosis [16]. It exerts an adverse action on autophagy-dysfunctional endothelial cells, inducing cell death with progression of atherosclerosis [16]. In the atherosclerotic aortic tissue of mice, CRAMP was detected in neutrophils [17]. Furthermore, CRAMP promotes atherosclerosis by enhancement of inflammatory monocyte recruitment [17]. The role of CAMP/CRAMP in the development of CAD and its clinical relevance has not been elucidated satisfactorily so far.

The present study aimed to investigate levels of CAMP/CRAMP, an antimicrobial peptide regulated by inflammatory and metabolic parameters, in the context of atherosclerosis, a disease caused by inflammatory and metabolic factors. The present study analyzed CRAMP serum levels in murine models of atherosclerosis and myocardial infarction. In addition, *Cramp* gene expression was determined in various organs and tissues of wild-type mice and low-density lipoprotein receptor deficiency (Ldlr^−/−^) mice on a high-fat diet compared to a standard diet in order to investigate the main site of CRAMP production under normo- and hypercholesterolemic conditions. The main focus of the study was the translational approach investigating CAMP serum levels and their correlations with metabolic and pathophysiological parameters in a large and well-characterized cohort of patients with acute and chronic CAD.

## 2. Results

### 2.1. Increased CRAMP Serum Levels in Atherosclerotic Mice

Obesity and the accompanied state of low-grade inflammation are one of the main risk factors for atherosclerosis [18]. As an antimicrobial peptide secreted by adipocytes and adipose tissue, CAMP represents one of the inflammatory parameters in metabolic dysfunction [19]. CAMP is also produced in atherosclerotic lesions, where it may enhance the innate immune response in atherosclerosis [20]. As atherosclerosis is an inflammatory disease aggravated by visceral obesity, dysregulation of adipokines in obesity is considered a putative underlying mechanism in atherogenesis. First, we analyzed CRAMP serum levels in Ldlr^−/−^ mice, a well-established mouse model, to study experimental atherosclerosis. Male mice at 10-weeks of age either continued to receive a standard diet (SD, *n* = 7) or were given an atherosclerosis-inducing high-fat diet (HFD, *n* = 12) for 12 weeks. The induction of atherosclerosis was confirmed by Oil red O staining in serial sections of the aortic root. In blood serum gained post-mortem, CRAMP levels were significantly increased in atherosclerotic animals (*p* < 0.001) (Figure 1).

### 2.2. Elevated Cramp Gene Expression in the Spleen and Liver of Atherosclerotic Mice

Cathelicidins are expressed and secreted by various cell types [21]. As we detected increased circulating CRAMP levels in atherosclerotic mice, we then investigated the source of increased CRAMP release. Therefore, we conducted gene expression analyses in different organs and tissues of atherosclerotic Ldlr^−/−^ mice. *Cramp* gene expression was massively increased in atherosclerotic Ldlr^−/−^ mice fed a HFD when compared to control Ldlr^−/−^ mice fed a SD in the spleen and liver, suggesting these organs as the potential source of elevated circulating CRAMP levels. In adipose tissue, kidney, colon, and ileum, *Cramp* gene expression was not modified by the HFD. *Cramp* gene expression could not be detected in aortic tissue (Figure 2).

In order to gain further insight into the general tissue expression of *Cramp*, we determined *Cramp* gene expression in numerous organs, tissues and glands in C57BL/6 wild-type mice. We detected moderate *Cramp* gene expression in most samples, with strong expression in the epididymal adipose tissue, spleen, thymus and aortic lymph node, and strongest expression in the bone marrow. Interestingly, we found no *Cramp* gene expression in the skin and bladder (Figure 3).

### 2.3. Decreased CRAMP Serum Levels after Experimental Myocardial Infarction in Mice

Progressive atherosclerosis frequently leads to plaque growth and plaque rupture in coronary arteries followed by occlusion of coronary arteries subsequently leading to myocardial infarction with cardiac ischemia [3]. Since systemic CRAMP is substantially regulated and might play a major role in atherosclerosis, we investigated CRAMP serum levels in two different models of experimental myocardial infarction by left anterior descending (LAD)-ligation. Short-term (by transient LAD occlusion, TO) as well as persistent (by permanent LAD occlusion, PO) myocardial ischemia significantly reduced CRAMP serum levels in C57BL/6J wild-type mice (Figure 4).

### 2.4. Circulating CAMP Levels in CAD Patients

Since we could demonstrate a significant influence of HFD-induced atherosclerosis and myocardial infarction on circulating CRAMP levels in the experimental mouse models, we investigated CAMP serum levels in CAD patients in a translational approach. Our main focus was on the comparison of circulating CAMP levels in a cohort of patients with different degrees of CAD. 234 participants were enrolled in this study. Patients were summarized in subgroups regarding their history of heart disease. In total, there were more male than female participants (177 male and 57 female). Patients with cardiovascular disease were 60.96 ± 7.53 years old, whereas controls were 54.10 ± 6.04 years old, i.e., slightly younger. CAMP serum levels were 87.06 ± 40.14 pg/mL in the entire study cohort but did not differ significantly between the subgroups (Figure 5).

Most CAD patients have comorbidities, such as arterial hypertension, dyslipidaemia, or diabetes mellitus, and were treated accordingly (antiplatelets, statin, β-blocker, angiotensin-converting enzyme inhibitor, angiotensin II receptor blocker, etc.). The basic characteristics of the patients are listed in Table 1. It is known that CAMP is produced and secreted by monocytes, which—like macrophages—are involved in the development of atherosclerosis. In our study, we therefore categorized the monocytes into their subgroups, i.e., classical monocytes (CD14^++^/CD16^−^), intermediate monocytes (CD14^++^/CD16^+^), and non-classical monocytes (CD14^+^/CD16^++^).

### 2.5. Correlation Analyses

In order to check whether the CAMP levels are related to one of the assessed basic parameters collected, we carried out extensive correlation analyses. In the entire CAD cohort, CAMP serum levels were positively correlated with triglycerides (*p* = 0.028, rho = 0.176, *n* = 156) (Figure 6A), especially in male CAD patients (*p* = 0.026, rho = 0.200, *n* = 125). Of note, the positive correlation was even stronger in CCS patients (*p* < 0.001, rho = 0.514, *n* = 41) (Figure 6B). There was no significant correlation of CAMP levels with other lipid parameters, although CAMP serum levels were significantly correlated with weight in women (*p* = 0.017, rho = 0.315, *n* = 57) (Figure 6C). In the entire CAD patient collective, CAMP levels were positively correlated with diastolic blood pressure (*p* = 0.014, rho = 0.182, *n* = 180) (Figure 6D) as well as with systolic blood pressure (*p* = 0.037, rho = 0.155, *n* = 180) (Figure 6E). Interestingly, CAMP serum levels were negatively correlated with left ventricle ejection fraction in patients with recurrent acute events (*p* = 0.026, rho = −0.376, *n* = 35) (Figure 6F). An overview of all performed correlation analyses is displayed in Appendix A. Furthermore, CAMP serum levels differed significantly in female patients depending on their NYHA (New York Heart Association) classification (*p* = 0.021). Regarding the total study cohort, no significant difference in CAMP serum levels was detected for subgroups defined by sex, BMI, dyslipidaemia, family history of CAD, or smoking (Appendix A).

In male CAD patients, CAMP serum levels were positively correlated with the number of circulating leukocytes (*p* = 0.047, rho = 0.163, *n* = 149) (Figure 7A). CAMP serum levels were negatively correlated with the percentage of CD14^++^/CD16^−^ monocytes (*p* = 0.010, rho = −0.380, *n* = 45) (Figure 7B) in a sub-cohort of lean patients (BMI < 25 kg/m^2^) and in women (*p* = 0.038, rho = −0.292, *n* = 51), whereas no significant correlation with further circulating monocyte subpopulations was detected. Within the entire study cohort, CAMP serum levels were positively correlated with MCP-1 levels (*p* = 0.026, rho = 0.239, *n* = 86) (Figure 7C), whereas a negative correlation was observed in ACS-1 patients (*p* = 0.030, rho = −0.512, *n* = 18). CAMP serum levels were positively correlated with systemic IL-1β in the control group (*p* = 0.041, rho = 0.461, *n* = 20) and in the subgroup of ACS-2 patients with recurrent acute events (*p* = 0.033, rho = 0.446, *n* = 23) (Figure 7D).

## 3. Discussion

Obesity is a strong risk factor for cardiovascular diseases [22]. In obesity, adipocytes and adipose tissue produce predominant proinflammatory adipokines [4]. The state of local and systemic low-grade inflammation is aggravated by an increased amount and activation/polarization of macrophages in adipose tissue [23,24]. As we described previously, nutritional carbohydrates and lipids can regulate postprandial CAMP serum levels in humans [24,25]. Ldlr^−/−^ mice are well established as a model for nutritional-induced atherosclerosis. Thus, for the first time, we quantified CRAMP serum levels in atherosclerotic Ldlr^−/−^ mice. We first investigated circulating CRAMP levels in Ldlr^−/−^ mice fed a HFD showing substantial arterial plaque burden. They had significantly increased serum CRAMP levels compared to mice fed a standard diet. Therefore, diet-induced atherosclerosis appears to favor increased CRAMP serum levels.

Cathelicidins are produced by various cell types and organs and are secreted by monocytes, keratinocytes, natural killer cells, lymphocytes, adipocytes, etc. [11]. Since cathelicidin has never been investigated in atherosclerotic Ldlr^−/−^ mice before, we aimed to elucidate the potential source of the increased CRAMP serum levels. Therefore, we analyzed *Cramp* gene expression in different organs and tissues, such as the aorta, spleen, kidney, liver, intestines, and adipose tissue. Interestingly, we found a significant increase in *Cramp* gene expression in the spleen and the liver in Ldlr^−/−^ mice fed an atherosclerosis-inducing HFD compared to Ldlr^−/−^ mice fed a SD. Contrary to previous studies reporting *Cramp* gene expression in atherosclerotic plaques [26], *Cramp* gene expression was not detectable in Ldlr^−/−^ murine aortic tissue in our study. In the other organs analyzed, *Cramp* gene expression was not altered by HFD. Since the spleen and liver are important immunological organs, our findings suggest that cathelicidins represent a further link between metabolism and inflammation. Increased CRAMP serum levels and increased *Cramp* gene expression in the liver and spleen could be due to direct nutritional effects of HFD, as well as immunological processes during atherogenesis. Therefore, future studies are needed to clarify the specific contributions of diet and vascular inflammation.

Elevated circulating CRAMP levels in Ldlr^−/−^ mice may indicate that cathelicidins play a role in atherosclerosis. The state of low-grade inflammation in obesity enhances atherosclerosis [18], and inflammatory mediators in obesity, such as TNF-α and cfDNA, affect adipocytic *Cramp* gene expression [27]. Cathelicidins are considered to represent a putative molecular mechanism in adipose tissue innate immunity [27], and therefore might be involved in the development of atherosclerosis. As atherosclerosis progresses, stenosis leads to narrowing of the coronary arteries, followed by occlusion due to progressive plaque growth or plaque rupturing [3]. Our experimental murine models demonstrated that both transient and permanent occlusion of one of the major coronary arteries (LAD)—and resulting transient or persistent myocardial ischemia—resulted in a decline of CRAMP serum levels in mice.

Mohanty et al. demonstrated that activation of hypoxia-inducible factor 1 (HIF-1) enhanced *Cramp* gene expression [28]. Furthermore, higher levels of proinflammatory cytokines and CAMP correlate with HIF-1 in type 2 diabetic patients [28]. Since myocardial infarction results in hypoxemia [3], hypoxic effects might also play a role in modulating circulating CAMP/CRAMP levels in our experimental models. To summarize the animal models, CRAMP levels are increased in chronic vascular inflammation/atherosclerosis after 12 weeks of HFD, whereas they are decreased by ischemia after LAD ligation. However, blood sampling after LAD ligation was performed after 28 days when the hearts were sampled for morphometric analyses. It would be interesting to analyze the CRAMP levels in both models at earlier time points. In other words, during the early phase of atherosclerosis after 4–6 weeks and 1–3 days after LAD ligation.

Based on these intriguing findings in mice, the potential implications and the fact that CAMP was detected in human atherosclerotic lesions post-mortem [26], we analyzed human CAMP serum levels with a translational approach. In the present study, we included patients with either CCS, ACS (while having a first acute event), ACS (while having a recurrent acute event), and controls without any history of cardiovascular diseases. In contrast to our findings in the murine models, we detected no significant difference in CAMP serum levels between these subgroups. Nonetheless, in a cohort of Asian patients, systemic CAMP levels in patients with STEMI were decreased compared with no CAD [29]. Recently in 2022, Zhao et al. discussed high basal plasma levels of CAMP as a potential predictor of 3-year risks of ischemic cardiovascular events in patients after STEMI [30]. Elevated cardiovascular risk is associated with dyslipidaemia [3]. Therefore, we conducted correlation analyses with CAMP. Of note, in the entire study cohort, CAMP serum levels positively correlated with triglycerides. In CCS patients, the positive correlation between CAMP serum levels and triglycerides was even more pronounced. In our previously published study, we were able to demonstrate a significant decrease in CAMP serum levels after oral lipid ingestion in healthy human individuals [25]. As the blood was not drawn from fasting patients, we cannot exclude a bias by the effects of nutritional lipids on systemic CAMP levels.

We observed statistical relations between CAMP serum levels and compromised myocardial function as a sign of heart failure, i.e., decreased left ventricular ejection fraction. Furthermore, CAMP serum levels differed depending on the status of NYHA classification in women. Therefore, CAMP might play a role in the development of heart failure. Interestingly, Zhou et al. found serum levels of CAMP to be significantly decreased in acute heart failure patients and negatively correlated with NT-proBNP, alongside anti-hypertrophic effects of CAMP in cardiomyocytes [31]. Investigating the possible role of CAMP in patients with heart failure therefore appears to be a promising goal for future research.

In male study participants, leukocytes were significantly positively correlated with CAMP serum levels. Of particular interest, we also observed a positive correlation of CAMP serum levels with proinflammatory cytokines like MCP-1 and IL-1β. These findings are in good accordance with previously described immunomodulatory properties of CAMP [32].

Our divergent results in mice and human patients suggest a species-dependent CAMP regulation in cardiovascular diseases. Activities of cathelicidins are modulated by complex interactions with the microenvironment, as well as the disease background [33]. Cathelicidins are produced by many cell types and regulated differentially in those cell types [33]. Furthermore, one should bear in mind that we identified the source of CRAMP in mice to be the immunologically active organs, such as the spleen and liver, and not in atherosclerotic vascular tissue. In contrast, CAMP was detected in human atherosclerotic lesions [26]. However, since cardiac ischemia was caused by external mechanical arterial occlusion in the applied mouse models, whereas pathophysiological myocardial ischemia is due to internal occlusion [3], contrary data from experimental animals with those from clinical cohorts should be interpreted with caution. Under pathophysiological conditions, myocardial infarction is caused by atherosclerotic plaque progression and thrombocyte activation [3]. Furthermore, in the mouse model, atherosclerosis was induced by a special cholesterol-rich diet. Therefore, potential dietary effects on circulating CAMP quantities have to be considered as well. In addition, patients are generally older, often have comorbidities and are treated with a variety of medications; none of this is the case in animal experiments.

Gene expression is controlled by regulatory promoter elements for transcription factors and miRNAs, among other things. However, there is still no comparative functional promoter study to clarify whether the species-dependent differences found in our study could be due to different promoter regulations. CAMP and CRAMP have an amino acid identity of ~55% and show partial differences in sequence and length, which also suggests species-dependent differences in protein function.

In summary, we demonstrate that CAMP/CRAMP appears to be regulated differently in experimental animal models and CAD patients. In mice, we were able to demonstrate that a high-fat diet causing atherosclerosis results in increased systemic levels of CAMP and an increased *Cramp* gene expression in the liver and spleen. Therefore, the antimicrobial peptide cathelicidin might be one of the missing key points in comprehending the complex interrelation of metabolism and inflammation in the development of atherosclerosis. Our findings on the regulation of CRAMP in experimental animal models may provide a basis for further mechanistic studies on the role of cathelicidins in atherosclerosis and myocardial infarction in these models. In contrast to these findings, no regulation of CAMP in CAD patients is observed, with its potential role as a biomarker in cardiovascular diseases thus remaining rather questionable. However, there are interesting correlations between risk factors and inflammatory parameters requiring further analysis of CAMP in CAD patients.

## 4. Materials and Methods

### 4.1. Animal Experiments

The animal handling and all experimental procedures were in accordance with the guidelines from directive 2010/63/EU of the European Parliament on the protection of animals used for scientific purposes, and they were also approved by the local animal care and use committee (50/2015, B2/1228).

Experimental atherosclerosis: Low-density lipoprotein receptor-deficient (Ldlr^−/−^, B6.129S7-*Ldlr^tm1Her^*/J) male mice at the age of 10 weeks were either continued on a standard diet (SD) or a fed a high-fat diet (HFD, D12079B, containing 21% fat and 0.21% cholesterol, Research Diets, New Brunswick, NJ, USA) for 12 weeks. At the end of this period, blood samples were collected after 4 h of starvation, and organs and tissues were isolated.

Surgical procedures of myocardial infarction: Male C57BL/6J wild-type mice at the age of 10–12 weeks were subjected to permanent or transient ligation of the LAD. Mice were anesthetized with isoflurane (4%) and intubated using a 20-gauge intravenous catheter with a blunt end. Mice were artificially ventilated with a stroke volume of 1–1.5 mL and at a rate of 60–80 strokes per minute using a rodent ventilator with a mixture of O_2_ and air (80%), to which isoflurane (2.0%) was added. The mouse was placed on a heating pad to maintain its body temperature at 37 °C. The chest hair was removed, and the chest was opened in the third intercostal space. A Prolene suture (6-0) was used to ligate the left coronary artery. Infarction was confirmed by discoloration of the ventricle and ST-T changes on the electrocardiogram. For transient ligation, the suture was removed after 45 min and reperfusion was allowed. The chest and skin were closed with a 5-0 silk suture. The animals were extubated before they were allowed to recover from the surgery. Sham-operated mice served as controls. Blood was collected after 28 days.

Tissues and organs were isolated from male C57BL/6J wild-type mice at the age of ~10 weeks and were disrupted using stainless steel beads (5 mm, Qiagen, Hilden, Germany) for subsequent RNA isolation and real-time PCR. Isolated tissues and organs are listed in Figure 3.

### 4.2. Study Cohort

In this single-center study, 203 male and female patients undergoing coronary angiography for the diagnosis and percutaneous intervention of CAD were enrolled in the University Hospital of Giessen and Marburg (UKGM) in the Department of Cardiology, Angiology, and Intensive Care Medicine, as were 31 healthy controls without cardiovascular diseases. Patients received standard cardiovascular care and medication (ACE-inhibitor, AT_1_-receptor blocker, β-blocker, diuretics, statin) according to the current guidelines. We investigated patients with proven coronary artery disease by coronary angiography and stenosis larger than 70% in relevant vessels that require intervention. Patients were classified into 2 groups: (1) Chronic coronary syndrome (CCS) patients, in which CAD was diagnosed more than 6 months ago and who, in the meantime, have had no further signs or symptoms of coronary artery disease under appropriate medication (*n* = 64; 53 male and 11 female); (2) acute coronary syndrome (ACS) patients with first and acute ST-elevation myocardial infarction (STEMI), non-ST-elevation myocardial infarction (NSTEMI) or unstable angina, as diagnosed by electrocardiography (ACS-1 *n* = 73; 53 male and 20 female). A combination of the two patients groups described before was used, featuring patients who had a diagnosis of coronary artery disease with intervention 6 months or longer ago and developed a second event (recurrent acute event) of STEMI, NSTEMI, or unstable angina with need for intervention. (ACS-2, *n* = 66; 54 male and 12 female). Blood samples were taken from acute patients a maximum of 4 days after diagnosis/intervention. For some analyses, the subgroups CCS, ACS-1, and ACS-2 were summarized in the group labeled coronary artery disease (CAD, *n* = 203). EDTA blood was collected from each subject, and further processing was performed within 2 h. Based on a standardized questionnaire, 31 controls (17 male and 14 female) without any history of cardiovascular disease were included in the study. Exclusion criteria were malignant diseases and either missing written informed consent or an inability to comply and to understand the investigational nature of the study and participation in other interventional drug or treatment trials. The study itself was conducted in accordance with the guidelines of the Declaration of Helsinki, and the research protocol, including the case report forms, was approved by the local ethics committee (245-12). Written informed consent was obtained from all study participants. More detailed information on the study cohort was previously described in detail [34].

### 4.3. Isolation of mRNA and Quantitative Real-Time PCR Analysis of Cramp Gene Expression in Murine Tissues

For the analysis of mRNA expression, total RNA from different tissues of C57BL/6J wild-type and Ldlr^−/−^ mice was isolated using RNA-Solv^®^ Reagent (Omega Bio-tek, Norcross, GA, USA) following the manufacturer’s instructions and reverse-transcribed with SuperScript reverse transcriptase, oligo(dT) primers (Thermo Fisher Scientific, Waltham, MA, USA), and deoxynucleoside triphosphates (Promega, Mannheim, Germany). Real-time PCR was performed in duplicate in a total volume of 20 μL using Power SYBR green PCR master mixture (Thermo Fisher Scientific) or real-time PCR probes and TaqMan Fast Advanced MasterMix (Thermo Fisher Scientific) on a Step One Plus real-time PCR system (Applied Biosystems, Foster City, CA, USA) in 96-well PCR plates (Applied Biosystems). SYBR Green or FAM/BHQ-1 fluorescence emissions were monitored after each cycle. For normalization, expression of glyceraldehyde-3-phosphate dehydrogenase (*Gapdh*) as a housekeeping gene was determined in duplicate. Relative gene expression was calculated by using the 2^−ΔΔCt^ method. Real-time PCR primers and probes were obtained from Microsynth AG (Balgach, Switzerland) and are available upon request. Modification of real-time PCR probes: 5′ = FAM, 3′ = BHQ-1.

### 4.4. Quantification of CRAMP/CAMP in Murine and Human Blood via Enzyme-Linked Immunosorbent Assay (ELISA)

Serum levels of CRAMP/CAMP from C57BL/6J wild-type mice, Ldlr^−/−^ mice and humans were measured in technical duplicates by enzyme-linked immunosorbent assay (ELISA) (kit purchased from Abbexa Ltd., Cambridge, UK (mouse) and Hycultec, Beutelsbach, Germany (human)). The detection limit of the ELISA kit used was 1.56 ng/mL (mouse) and 0.14 ng/mL (human). All measurements exceeding an intra-duplicate variance of 20% were repeated.

### 4.5. Statistical Analysis

A statistical software package (SPSS 28.0.0) was used for calculation. Descriptive analyses are expressed as means ± standard deviation in the tables. Mean values were compared using the Mann–Whitney U-test for non-related samples and the Kruskal–Wallis test (>2 unrelated samples), while the Bonferroni method was used to correct for multiple testing. Correlation analysis was performed by applying the Spearman rho test for linear variables. Additionally, *p* values below 0.05 (two-tailed) were considered statistically significant. The results are visualized as box plots, giving the median and the quartile ranges. Dots mark statistical outliers.

## Figures and Tables

**Figure 1 ijms-25-02909-f001:**
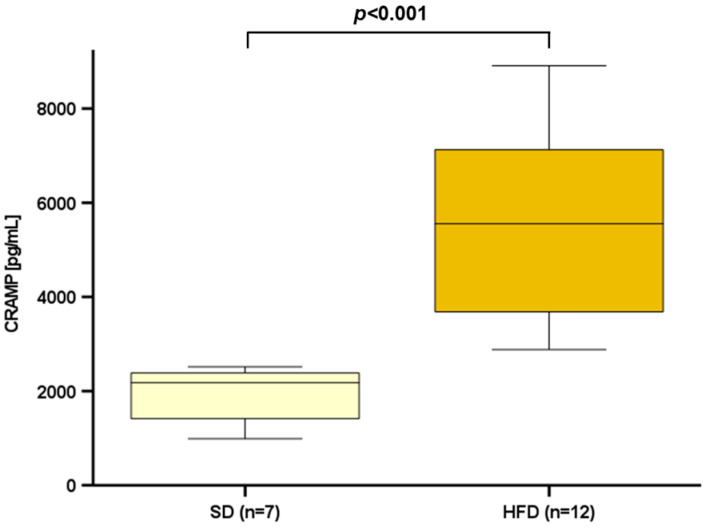
Serum CRAMP levels were significantly increased in Ldlr^−/−^ mice fed a high-fat diet (HFD) for 12 weeks compared to mice maintained on a standard diet (SD). Blood serum samples from Ldlr^−/−^ mice were collected at the age of 22 weeks and CRAMP levels were measured by ELISA. Mann–Whitney U-test was applied for the calculation of statistical significance. Samples from *n* = 7–12 animals per group were investigated.

**Figure 2 ijms-25-02909-f002:**
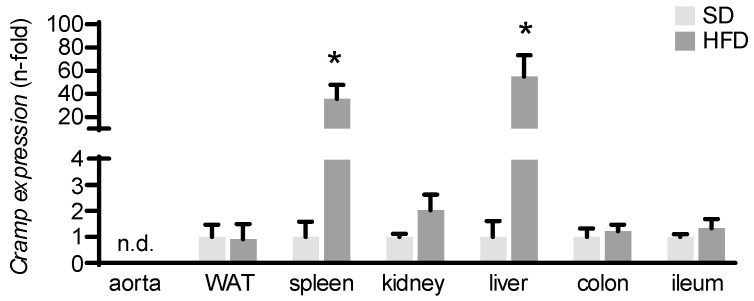
*Cramp* gene expression was significantly induced in the spleen and liver of Ldlr^−/−^ mice fed a HFD. Tissues were collected from mice after 12 weeks of a HFD or a continued SD at the age of 22 weeks. After RNA isolation, *Cramp* gene expression was measured by RT-PCR. Kruskal–Wallis test was applied for the calculation of statistical significance. Samples from *n* = 4–5 animals per group were investigated. * *p* < 0.05 vs. SD. n.d.: not detectable.

**Figure 3 ijms-25-02909-f003:**
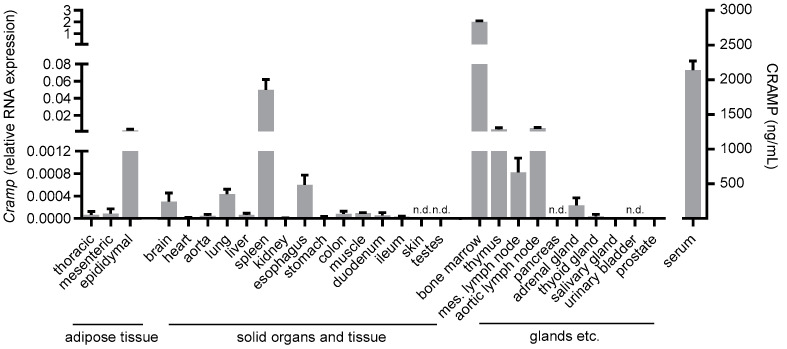
Tissue gene expression of *Cramp*. After RNA isolation, CRAMP mRNA levels in different murine tissues and organs were determined by RT-PCR, and CRAMP serum levels were quantified by ELISA. Samples from *n* = 3–5 animals were investigated for gene expression and *n* = 10 for serum levels. n.d.: not detectable.

**Figure 4 ijms-25-02909-f004:**
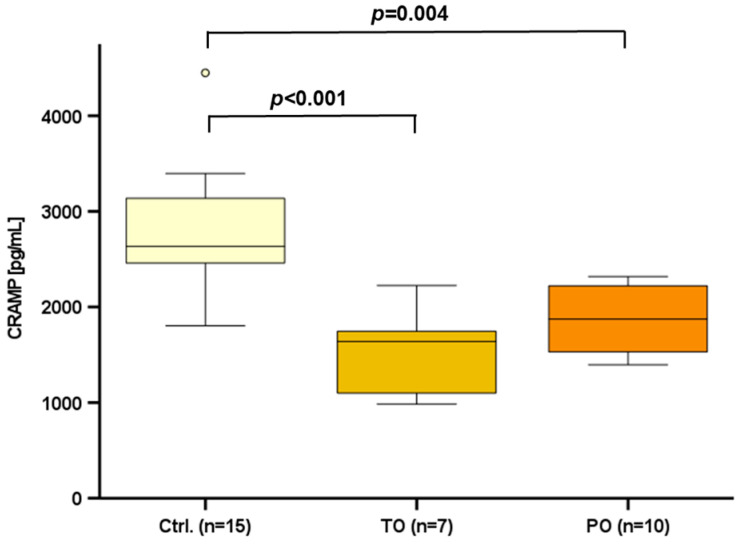
CRAMP serum levels were significantly diminished after transient occlusion (TO) and permanent occlusion (PO) of the LAD in wild-type mice. Blood serum samples were collected at the age of 10–12 weeks and CRAMP levels were measured by ELISA. Kruskal–Wallis-test was applied for the calculation of statistical significance. Samples from *n* = 7–15 animals per group were investigated. Dots mark statistical outliers.

**Figure 5 ijms-25-02909-f005:**
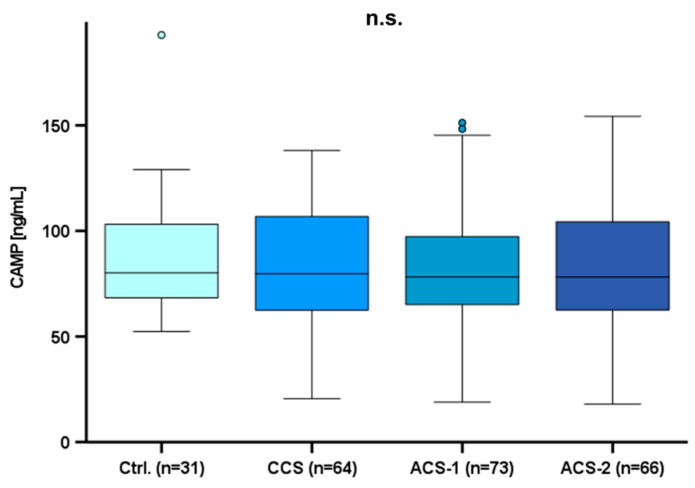
CAMP serum levels did not differ between subgroups. Circulating CAMP levels were quantified by ELISA. The Kruskal–Wallis test was applied for the calculation of statistical significance. Dots mark statistical outliers. Outliers > 2.5 SD are not shown; n.s.: not significant; CAMP: Cathelicidin antimicrobial peptide; Ctrl.: Control; CCS: chronic coronary syndrome; ACS: acute coronary syndrome.

**Figure 6 ijms-25-02909-f006:**
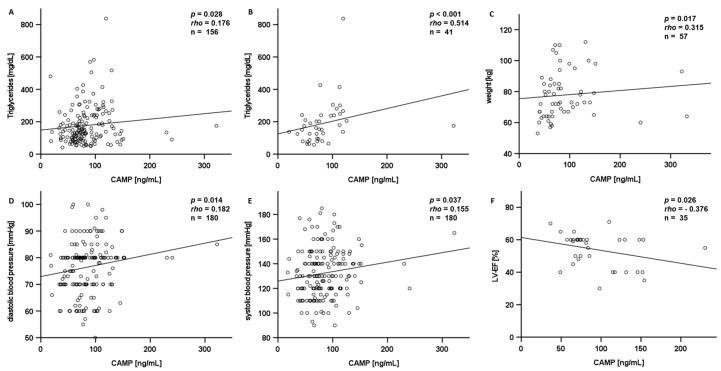
Circulating CAMP levels were positively correlated with triglyceride levels in CAD patients (**A**), especially in the subgroup of CCS patients (**B**). In women, weight and CAMP levels were positively correlated (**C**). CAMP serum levels were positively correlated with diastolic (**D**) and systolic (**E**) blood pressure in CAD patients. CAMP serum levels were negatively correlated with left ventricle ejection fraction in patients with recurrent acute events (**F**). CAMP serum levels were measured by ELISA. The Spearman-rho test was applied for the calculation of *p* values and statistical significance. CAMP: Cathelicidin antimicrobial peptide; LV-EF: left ventricular ejection fraction.

**Figure 7 ijms-25-02909-f007:**
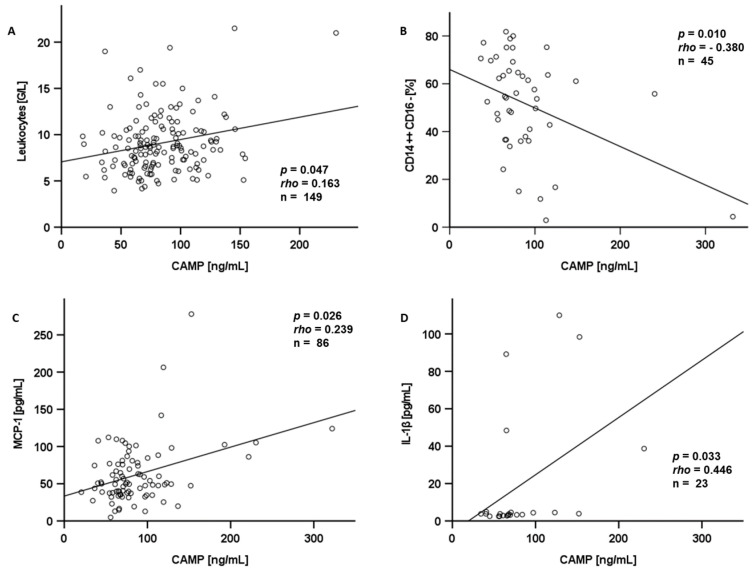
Circulating CAMP levels were correlated with proinflammatory parameters. Leukocytes in male CAD patients (**A**), MCP-1 in the entire study cohort (**C**), and IL-1β in ACS-2 (**D**) were correlated positively with circulating CAMP levels. The amount of CD14^++^CD16^−^ monocytes in lean patients was correlated negatively with circulating CAMP levels (**B**). CAMP serum levels were measured by ELISA. The Spearman-rho test was applied for the calculation of *p* values and statistical significance. CAMP: Cathelicidin antimicrobial peptide; MCP-1: monocyte chemoattractant protein-1; IL-1β: Interleukin-1β.

**Table 1 ijms-25-02909-t001:** Basic characteristics of the study population.

	Ctrl. (*n* = 31)	CCS (*n* = 64)	ACS-1 (*n* = 73)	ACS-2 (*n* = 66)
sex (men/women)	17/14	53/11	53/20	54/12
age [y]	54.10 ± 6.04	63.41 ± 7.86 *	57.93 ± 5.80 °	61.92 ± 7.86 *^+^
BMI [kg/m^2^]	27.97 ± 4.77	30.57 ± 5.43	28.39 ± 4.57	29.13 ± 5.39
Total cholesterol [mg/dL]	n.a.	174.33 ± 45.80	209.14 ± 57.43 °	176.96 ± 46.39 ^+^
LDL cholesterol [mg/dL]	n.a.	113.80 ± 34.48	144.57 ± 56.57 °	112.57 ± 34.36 ^+^
HDL cholesterol [mg/dL]	n.a.	49.07 ± 10.49	51.93 ± 16.78	50.85 ± 17.45
Triglycerides [mg/dL]	n.a.	190.46 ± 136.94	158.75 ± 98.71	190.98 ± 131.55
Leukocytes [giga/L]	n.a.	7.70 ± 2.06	9.84 ± 3.60 °	8.99 ± 2.91
Monocytes [% total monocytes]	54.09 ± 22.69	64.28 ± 19.70	55.73 ± 25.31	65.83 ± 16.64
CD14^++^/CD16^−^ [% total monocytes]	77.81 ± 20.22	82.09 ± 13.82	80.90 ± 15.95	84.53 ± 12.74
CD14^++^/CD16^+^ [% total monocytes]	7.03 ± 5.07	8.42 ± 10.19	9.47 ± 8.08	7.96 ± 5.86
CD14^+^/CD16^++^ [% total monocytes]	18.02 ± 25.74	10.20 ± 6.86	10.70 ± 13.25 °	7.65 ± 9.02 °
CRP [mg/L]	n.a.	31.11 ± 34.29	45.17 ± 57.97	19.28 ± 25.24
CAMP [ng/mL]	97.94 ± 57.46	86.47 ± 40.61	84.23 ± 33.05	85.66 ± 37.08
MCP-1 [pg/mL]	76.13 ± 25.39	64.09 ± 40.95	56.87 ± 23.96	51.25 ± 55.94 *
IL-1β [pg/mL]	11.64 ± 16.76	3.65 ± 0.88	4.33 ± 1.76	19.49 ± 33.76

Mean values ± standard deviation are presented. * *p* < 0.05 versus Ctrl.; ° *p* < 0.05 versus CCS; ^+^
*p* < 0.05 versus ACS-1; n.a.: not available; CAMP: Cathelicidin antimicrobial peptide; Ctrl.: Control; CCS: chronic coronary syndrome; ACS: acute coronary syndrome; BMI: body mass index; LDL: low-density lipoprotein; HDL: high-density lipoprotein; CRP: C-reactive peptide; MCP-1: monocyte chemoattractant protein-1; IL-1β: Interleukin-1β.

## Data Availability

The data presented in this study are available on reasonable request from the corresponding author.

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
