# Peer review of "Cathelicidin Antimicrobial Peptide Levels in Atherosclerosis and Myocardial Infarction in Mice and Human"

_ijms, 2024, doi:10.3390/ijms25052909_

Round 1
Reviewer 1 Report
Comments and Suggestions for Authors
Review article IJMS-2871545
The submitted article delves into cathelicidin research within the context of atherosclerosis. Regrettably, I find it challenging to provide a more detailed summary as I couldn't discern a clear overarching goal in the paper. It gives me the impression that the entire work, including the results presented, appears to be the outcome of experiments without a deeper conceptual foundation for its design and execution. Below, I'll elaborate on specific comments.
ABSTRACT: Abstract is too long and have too much details. It should get only type of the experiments.
Line 15 coincidence - it is too colloquial word in a scientific text.
Line 17 What’s mean polypeptide (definition) ? it should be only "peptide".
Line 27-30 too much details
Line 56 "Cathelicidin antimicrobial peptide (CAMP, also known as LL-37)": I have big problem with this phrase. Because LL-37 is HUMAN cathelicidin , so here it should be necessarily be specified. If CAMP is the same as LL37, how did you label this peptide in mice, given that it is only found in humans? I am aware that there is a mouse equivalent of this peptide, but it has a slightly different sequence and, most importantly, a different length (it is two amino acid residues shorter at the
N-terminus). This is why I think that clarification in the introduction is necessary, especially since the first three studies are conducted on mice and the next one on humans, patients. Therefore, the manuscript must include information and a comparison of these peptides, and in specific experiments, it should be specified which one was labeled. Of course, there must also be information stating that they are analogues, so that a meaningful comparison between experiments can be made.
Line 116 C57BL/6 Why in Camp gene expression You used different kind of the mice?
Line 130 add TO and in the second bracket, PO like in the figure 4.
Line 131 wild-type mice: because You used two type of mice, here should be given a particular strain of mice
Line 232 Ldlr−/− : in the Results is the notification that in this experiment were wild type mice BalbC. So which mice were finally studied?
Line 252-255, Discussion: To be tempted to draw such a conclusion in the discussion, one should check the correlation between cortisol and cAMP concentrations oneself. As it stands, this is just far-fetched speculation without evidence. There are many such formulations in the discussion. Most of this discussion consists of literature facts, and far less here is a factual discussion of the results obtained. This is probably not the purpose of experimental work. When planning experiments, we as scientists must know very well why we are planning them and what their purpose is. Conducting experiments and presenting results solely to draw conclusions, without comparing them with anything, as the authors called 'may lead to multiple theories,' is pointless. I believe the broader context of the research is missing throughout the paper and should be added to publish this work.
Line 304: what is CVD ?? And why You conclude about this, if in the introduction ad in whole manuscript was CAD? This paragraph is weird, it doesn't quite fit me with this work. It is another example of far-fetched insinuations without supporting results.
Line 320-322: What happened with the blood samples and with organs and tissues? how the CAMP level _Results 2.1 and 2.3) , were done? In whole MM part we don't have information . details about experiments from part 2.1 to part 2.4)
Line 341 Study Cohort: Are You sure that is correct description? Because in part 2.4 you have sentence " In total, there were more 145 male than female participants (177 male and 57 female)" 177 + 57 = 234 patient, so for me it two different things
Line 273 4.3. Isolation of mRNA and Quantitative Real-Time PCR Analysis of CAMP Gene Expression in murine Tissues: in which experiment You isolated mRNA?
Line 460: the year and doi number are missing
Author Response
Response to Reviewer 1:
We thank Reviewer 1 for the valuable suggestions in order to improve the quality of our manuscript. Please find below our detailed response and the corrections that have been made. All corrections are highlighted in red color in the revised version of our manuscript.
The submitted article delves into cathelicidin research within the context of atherosclerosis. Regrettably, I find it challenging to provide a more detailed summary as I couldn't discern a clear overarching goal in the paper. It gives me the impression that the entire work, including the results presented, appears to be the outcome of experiments without a deeper conceptual foundation for its design and execution. Below, I'll elaborate on specific comments.
Reviewer: ABSTRACT: Abstract is too long and have too much details. It should get only type of the experiments.
Authors: Thank you for your suggestion to shorten the abstract. We have deleted details and shortened the abstract.
Reviewer: Line 15 coincidence - it is too colloquial word in a scientific text.
Authors: We have deleted coincidence in this context (p. 1, l. 15).
Reviewer: Line 17 What’s mean polypeptide (definition)? it should be only "peptide".
Authors: We thank you for this hint. Polypeptide refers to the length of the peptide (in contrast to oligopeptide) but is irrelevant here and we have therefore deleted it as suggested (p.1, l. 16).
Reviewer: Line 27-30 too much details
Authors: In order to shorten the abstract as recommended, we have removed details here (p.1, ll. 13-41).
Reviewer: Line 56 "Cathelicidin antimicrobial peptide (CAMP, also known as LL-37)": I have big problem with this phrase. Because LL-37 is HUMAN cathelicidin , so here it should be necessarily be specified. If CAMP is the same as LL37, how did you label this peptide in mice, given that it is only found in humans? I am aware that there is a mouse equivalent of this peptide, but it has a slightly different sequence and, most importantly, a different length (it is two amino acid residues shorter at the N-terminus). This is why I think that clarification in the introduction is necessary, especially since the first three studies are conducted on mice and the next one on humans, patients. Therefore, the manuscript must include information and a comparison of these peptides, and in specific experiments, it should be specified which one was labeled. Of course, there must also be information stating that they are analogues, so that a meaningful comparison between experiments can be made.
Authors: Thank you very much for this important hint, this was imprecise on our part. We added information in the abstract (p.1, ll. 20-21) and a section in the introduction (p. 2, ll. 56-62) and discussion (p.9, ll. 359-363) on that issue. As you noted rightfully, the term LL-37 is used for human CAMP only. In mice, CAMP is named CRAMP. To make a clear distinction, we now use CAMP/Camp for human cathelicidin and CRAMP/Cramp for murine cathelicidin.
Reviewer: Line 116 C57BL/6 Why in Camp gene expression You used different kind of the mice?
Authors: We used 2 different mouse strains for the animal experiments. For the studies on experimental atherosclerosis, we used Ldlr-KO animals (Figure 1+2), which are required for the development of atherosclerotic plaques under HFD (Apoe-KO mice would be an alternative strain for these experiments). In contrast, we used the common wild-type strain C57BL/6J from Jackson Laboratory to study Cramp expression in various organs/tissues (Figure 3) and also used the C57BL/6J wild-type strain for experimental infarction studies (Figure 4). We have described this in detail in the Material and Methods section (4.1. Animal experiments) (pp.9-10, ll. 379-406).
Reviewer: Line 130 add TO and in the second bracket, PO like in the figure 4.
Authors: We have added TO and PO at the appropriate place (p. 4, ll. 155-156).
Reviewer: Line 131 wild-type mice: because You used two type of mice, here should be given a particular strain of mice
Authors: We have added the strain C57BL/6J. To better distinguish experiments with wild-type mice from experiments with Ldlr-KO mice, we have now always written C57BL/6J wild-type mice throughout the manuscript (e.g. p. 3, l. 134; p. 4, l. 157).
Reviewer: Line 232 Ldlr−/− : in the Results is the notification that in this experiment were wild type mice BalbC. So which mice were finally studied?
Authors: As described above, we used two different strains. Ldlr-KO mice for the atherosclerosis studies and C57BL/6J wild-type mice for the expression studies in organs/tissues and the infarction studies. We did not use BALB/c mice in our experiments because the Ldlr-KO (B6.129S7-Ldlrtm1Her/J) strain was backcrossed to C57BL/6J and the experiments are therefore more comparable with C57BL/6J wild-type mice. Here (p. 3, ll. 116-131) we refer to increased Cramp expression in the spleen and liver of atherosclerotic animals and these experiments were therefore performed in Ldlr-KO mice.
Reviewer: Line 252-255, Discussion: To be tempted to draw such a conclusion in the discussion, one should check the correlation between cortisol and cAMP concentrations oneself. As it stands, this is just far-fetched speculation without evidence. There are many such formulations in the discussion. Most of this discussion consists of literature facts, and far less here is a factual discussion of the results obtained. This is probably not the purpose of experimental work. When planning experiments, we as scientists must know very well why we are planning them and what their purpose is. Conducting experiments and presenting results solely to draw conclusions, without comparing them with anything, as the authors called 'may lead to multiple theories,' is pointless. I believe the broader context of the research is missing throughout the paper and should be added to publish this work.
Authors: We want to thank the reviewer for this comment. Unfortunately, we are not able to quantify cortisol retrospectively in our study cohort. Cortisol operates as an immunosuppressive in many circumstances. CAMP, as an important peptide in inflammation, might be influenced by cortisol. Nonetheless, we are not able to prove this theory in the present study. We have therefore deleted this section, but plan to include cortisone as a routine parameter in our patient samples for future studies. Furthermore, we have revised the manuscript according to the reviewer's advice.
Reviewer: Line 304: what is CVD ?? And why You conclude about this, if in the introduction ad in whole manuscript was CAD? This paragraph is weird, it doesn't quite fit me with this work. It is another example of far-fetched insinuations without supporting results.
Authors: Thank you for this hint. CVD is an abbreviation for cardiovascular disease and was used as a synonym for CAD (coronary artery disease). As you mentioned rightfully, we should not use both terms in one manuscript. Therefore, we corrected “CVD” to “CAD” (p. 9, ll. 365, 373, 376).
Reviewer: Line 320-322: What happened with the blood samples and with organs and tissues? how the CAMP level _Results 2.1 and 2.3) , were done? In whole MM part we don't have information . details about experiments from part 2.1 to part 2.4)
Authors: Please note that all experiments in the results section are described in detail in the material and methods section. Please see under:
4.1. Animal experiments (HFD feeding, myocardial infarction surgery, tissue collection, etc.) (pp. 9-10, ll. 378-406)
4.2. Study Cohort (patients, medication, blood sampling, etc.) (p. 10, ll. 407-439)
4.3. Isolation of mRNA and Quantitative Real-Time PCR Analysis of CAMP Gene Expression in murine Tissues (for experiments in C57BL/6J wild-type mice and in Ldlr-KO mice) (pp. 10-11, ll. 440-457)
4.4. Quantification of CRAMP/CAMP in Murine and Human Blood via Enzyme-linked Immunosorbent Assay (ELISA) (Description of the ELISA measurements in the animal experiments and the patient cohort) (p. 11, ll. 458-466)
Reviewer: Line 341 Study Cohort: Are You sure that is correct description? Because in part 2.4 you have sentence " In total, there were more 145 male than female participants (177 male and 57 female)" 177 + 57 = 234 patient, so for me it two different things
Authors: Thank you very much for your thorough perusal of our manuscript. In this line, there was a typo and it should have been written “203” patients suffering from CAD (+31 controls = 234 in total, of which 177 male and 57 female). Further below the exact gender distribution in the subgroups is given. We have carefully checked all the patient numbers given and corrected this error (p. 10, l. 408).
Reviewer: Line 273 4.3. Isolation of mRNA and Quantitative Real-Time PCR Analysis of CAMP Gene Expression in murine Tissues: in which experiment You isolated mRNA?
Authors: We have isolated RNA from murine tissue for the expression analyses by PCR. To make this clear, we have now indicated this in the legend of Figures 2 and 3 (p. 3, l. 129; p. 4, l. 145).
Reviewer: Line 460: the year and doi number are missing
Authors: Thank you for this hint. We corrected the missing doi number (p. 13, l. 538). Unfortunately, no doi numbers was available on PubMed for one more paper (p.13, ll. 558-559).
Reviewer 2 Report
Comments and Suggestions for Authors
This manuscript has attempted to describe potential correlations of cathelicidin AMP expressions in atherosclerosis and myocardial infarction in mice and human. AMPs display multiple activities including host defence and immunomodulation. Human and mice both contain a single cathelicidin AMP. Cathelicidin AMP in human, LL37, is known for skin defence and many other functions. Authors hypothesized expression level of CAMP in obese conditions. In animal models (mice study) this study finds high expression of CAMP in HFD mice. Whereas no such increase was observed in cohort study of patients. Although, in correlation analysis triglycerides level and CAMP were found to be correlated. The study does provide new information about CAMP and diseases. Although, exact molecular mechanism is not certain yet. The differences observed between mice and human are not clear. Authors should provide some explanation. I am not sure what was the rationale choosing cathelicidin AMP in the study. Can authors study defensins in their experiments? In other words, study design needs to be better explained. The big problem in human study is that the only blood samples were examined.
Author Response
We thank reviewer 2 for the valuable suggestions and corrections that helped us to improve the quality of our manuscript. Please find below our detailed response and the corrections that have been made. All corrections are highlighted in red color in the revised version of our manuscript.
This manuscript has attempted to describe potential correlations of cathelicidin AMP expressions in atherosclerosis and myocardial infarction in mice and human. AMPs display multiple activities including host defence and immunomodulation. Human and mice both contain a single cathelicidin AMP. Cathelicidin AMP in human, LL37, is known for skin defence and many other functions. Authors hypothesized expression level of CAMP in obese conditions. In animal models (mice study) this study finds high expression of CAMP in HFD mice. Whereas no such increase was observed in cohort study of patients. Although, in correlation analysis triglycerides level and CAMP were found to be correlated.
Reviewer: The differences observed between mice and human are not clear. Authors should provide some explanation.
Authors: One possible explanation may lie in the differences between the experimental models and the CAD patients. In addition to general species differences, experimental models attempt to depict the situation in patients but can naturally only approximate this. There are multiple reasons for this: patients are usually older, often have comorbidities and are treated with a wide variety of medications. None of this is the case in animal experiments.
Another reason could lie in CAMP itself. Partly differences in the sequence and length between humans and mice may also affect the protein function. In mice, the term cathelin-related antimicrobial peptide CRAMP is therefore introduced. To make it easier to distinguish between the two forms in our study, we now use CRAMP/Cramp in mice and CAMP/Camp in humans (see e.g. p. 2, ll. 58-61). Gene expression is regulated, among other things, by regulatory promoter elements for transcription factors and miRNAs. To our knowledge, there is no comparative functional promoter study (Camp/Cramp) that could explain our observed differences.
We have added this to the discussion section (p. 9, ll. 342-363).
Reviewer: I am not sure what was the rationale choosing cathelicidin AMP in the study. Can authors study defensins in their experiments? In other words, study design needs to be better explained.
Authors: Thank you for the valuable comments and suggestions on our manuscript. We elaborated the introduction and the discussion section to clarify this issue. We chose CAMP as a peptide of interest in cardiovascular disease because CAMP might play an important role in the interrelation between metabolism and inflammation. Cardiovascular diseases are thought to be caused by malnutrition and inflammatory stress on the endothelium. In 2015, Zhang et al. discovered CAMP to be produced by adipocytes. They showed that CAMP expression in adipocytes is increased on sight of infection. Expression of defensins was not increased in this study (ref #15, doi:10.1126/science.1260972.) Further studies revealed CAMP regulation by metabolic factors in vivo and in vitro. Therefore, we concentrated on CAMP in the current study to investigate its potential regulation in cardiovascular diseases. Defensins are certainly also interesting candidates in this context, but the limited residual volume of many of our samples does not allow us to analyze other factors.
To clarify the study design, we elaborated the introduction section (p. 2, ll. 75-77)
Reviewer: The big problem in human study is that the only blood samples were examined.
Author: As mentioned by the reviewer, most clinical studies only use blood as test material. Albeit in a completely different study population of obese patients undergoing bariatric gastric surgery, we have investigated Camp gene expression in subcutaneous and visceral adipose tissue (ref #19, doi: 10.1055/a-1323-3050). In the present study, however, only serum samples were available.
Round 2
Reviewer 1 Report
Comments and Suggestions for Authors
Thank you for incorporating my comments into the new version of the manuscript. I have a few more minor comments that do not detract from the value of the manuscript.
1. Abstract is still incompatible with the IJMS standard. In the template file is wrote:
“Abstract: A single paragraph of about 200 words maximum. For research articles, abstracts should give a pertinent overview of the work. We strongly encourage authors to use the following style of structured abstracts, but without headings: (1) Background: Place the question addressed in a broad context and highlight the purpose of the study; (2) Methods: briefly describe the main methods or treatments applied; (3) Results: summarize the article’s main findings; (4) Conclusions: indicate the main conclusions or interpretations. The abstract should be an objective representation of the article and it must not contain results that are not presented and substantiated in the main text and should not exaggerate the main conclusions.”
But it is decision for Editor
2. Line 59: it should be in increasing order and in single parenthesis, Like: [5, 7, 8, 9]
Author Response
We thank Reviewer 1 for the valuable suggestions in order to improve the quality of our manuscript. Please find below our detailed response and the corrections that have been made. All corrections are highlighted in blue color (red color: first revision) in the revised version of our manuscript.
Reviewer: Thank you for incorporating my comments into the new version of the manuscript. I have a few more minor comments that do not detract from the value of the manuscript.
- Abstract is still incompatible with the IJMS standard. In the template file is wrote:
“Abstract: A single paragraph of about 200 words maximum. For research articles, abstracts should give a pertinent overview of the work. We strongly encourage authors to use the following style of structured abstracts, but without headings: (1) Background: Place the question addressed in a broad context and highlight the purpose of the study; (2) Methods: briefly describe the main methods or treatments applied; (3) Results: summarize the article’s main findings; (4) Conclusions: indicate the main conclusions or interpretations. The abstract should be an objective representation of the article and it must not contain results that are not presented and substantiated in the main text and should not exaggerate the main conclusions.”
But it is decision for Editor
Author: Thank you very much for your helpful comment. We shortened the abstract as recommended. The new abstract contains 200 words as suggested. p1, ll. 14-30
Reviewer: 2. Line 59: it should be in increasing order and in single parenthesis, Like: [5, 7, 8, 9]
Author: Thank you for this hint. We corrected the order of references and used single parenthesis. p. 2, l. 48
Reviewer 2 Report
Comments and Suggestions for Authors
All comments are addressed.
Author Response
Reviewer: All comments are addressed.
Authors: We thank reviewer 2 for the valuable suggestions and corrections that helped us to improve the quality of our manuscript.